

# Possible heterogeneous hydroxymethanesulfonate (HMS) chemistry in northern China winter haze and implications for rapid sulfate formation

Shaojie Song[1*], Meng Gao[1], Weiqi Xu[2,3], Yele Sun[2,3,4*], Douglas R. Worsnop[5], John T. Jayne[5], Yuzhong Zhang[1], Lei Zhu[1], Mei Li[6,7], Zhen Zhou[6,7], Chunlei Cheng[6,7], Yibing Lv[8], Ying Wang[9], Wei Peng[9], Xiaobin Xu[9], Nan Lin[10], Yuxuan Wang[11], Shuxiao Wang[12], J. William Munger[1], Daniel Jacob[1], Michael B. McElroy[1*]

[1]School of Engineering and Applied Sciences, Harvard University, Cambridge, MA 02138, USA
[2]State Key Laboratory of Atmospheric Boundary Physics and Atmospheric Chemistry, Institute of Atmospheric Physics, Chinese Academy of Sciences, Beijing 100029, China
[3]College of Earth Sciences, University of Chinese Academy of Sciences, Beijing 100049, China
[4]Center for Excellence in Regional Atmospheric Environment, Institute of Urban Environment, Chinese Academy of Sciences, Xiamen 361021, China
[5]Aerodyne Research, Inc., Billerica, MA 01821, USA
[6]Institute of Mass Spectrometer and Atmospheric Environment, Jinan University, Guangzhou 510632, China
[7]Guangdong Provincial Engineering Research Center for Online Source Apportionment System of Air Pollution, Guangzhou 510632, China
[8]China National Environmental Monitoring Center, Beijing 100012, China
[9]State Key Laboratory of Severe Weather & Key Laboratory for Atmospheric Chemistry of CMA, Chinese Academy of Meteorological Sciences, Beijing 100081, China
[10]Department of Earth System Science, Tsinghua University, Beijing 100084, China
[11]Department of Earth and Atmospheric Sciences, University of Houston, Houston, TX 77004, USA
[12]School of Environment, Tsinghua University, Beijing 100084, China

*Correspondence to*: Shaojie Song (songs@seas.harvard.edu), Yele Sun (sunyele@mail.iap.ac.cn), Michael B. McElroy (mbm@seas.harvard.edu)

**Abstract.** Chemical mechanisms responsible for rapid sulfate production, an important driver of winter haze formation in northern China, remain unclear. Here, we propose a potentially important heterogeneous hydroxymethanesulfonate (HMS) chemical mechanism. Through analyzing field measurements with aerosol mass spectrometry, we show evidence for a possible significant existence in haze aerosols of organosulfur primarily as HMS, misidentified as sulfate in previous observations. We estimate that HMS can account for up to about one-third of the sulfate concentrations unexplained by current air quality models. In addition, HMS in the presence of hydroxyl radicals can trigger rapid sulfate production in aerosol water. Heterogeneous production of HMS by $SO_2$ and formaldehyde is favored under northern China winter haze conditions due to high aerosol water content, moderately acidic pH values, high gaseous precursor levels, and low temperature. These analyses identify an unappreciated importance of formaldehyde in secondary aerosol formation and calls for more research on sources and on the chemistry of formaldehyde in northern China winter.





## 1 Introduction

Severe haze episodes occur frequently in Beijing and throughout the North China Plain (NCP), especially in winter, posing substantial threats to public health (Ding et al., 2016; Fu and Chen, 2017; Gao et al., 2017). High concentrations of fine particles and reduced visibility are associated with stagnant meteorological conditions, i.e., shallow boundary layers, weak winds, and

high relative humidity (RH) (Wang et al., 2014a; Cai et al., 2017; Tie et al., 2017). Rapid formation of particulate sulfate is considered one of the key drivers of haze pollution for several reasons: sulfate is an important component of fine particles; it facilitates the partitioning of gaseous ammonia into the particle phase; and it enhances aerosol water uptake, changing the optical and chemical properties of aerosols (Guo et al., 2014; Huang et al., 2014). Sulfate is also known to impact climate and acid deposition (Charlson et al., 1992; Xie et al., 2015).

Most of the sulfate is of secondary origin, formed by oxidation of anthropogenic $SO_2$ (He et al., 2018). The exponential relationship between RH and the molar ratios of sulfate relative to $SO_2$, as observed during 2014 winter in Beijing (Fig. 1; see Methods) and in many previous studies (Sun et al., 2013; Zheng et al., 2015b; Wang et al., 2016), implies that heterogeneous chemistry (processes involving both gas and aerosol phases) plays an important role in production of sulfate. Indeed, air quality

model simulations fail to capture the rapid increase of sulfate from clean to haze periods when considering only the oxidation of $SO_2$ in the gas phase and in cloud/fog water, suggesting missing heterogeneous sources of sulfate (Wang et al., 2014b; Zheng et al., 2015a; Li et al., 2017a) (Fig. 1). Adding an apparent RH-dependent heterogeneous uptake for $SO_2$ on aerosols greatly reduces the negative bias in the modeled sulfate concentrations (see Methods) (Wang et al., 2014b; Zheng et al., 2015a).

Several heterogeneous reaction pathways have been proposed (He et al., 2014; Cheng et al., 2016; Wang et al., 2016; Li et al., 2017a; Hung et al., 2018; Qin et al., 2018; Yu et al., 2018), including oxidation of $SO_2$ in aerosol water (by $NO_2$, transition-metal-catalyzed $O_2$, or $H_2O_2$) and on aerosol surfaces (by $NO_2$ and/or $O_2$). Their relative importance for sulfate production in winter haze, however, is unknown due to uncertainties in relevant reaction rates and estimates for aerosol water pH values (most reaction pathways are pH-dependent) (Guo et al., 2017b; Liu et al., 2017a; Li et al., 2018b; Wang et al., 2018; Zhao et

al., 2018). For example, Wang et al. (2016; 2018) found in laboratory experiments that the rate for oxidation of $SO_2$ by $NO_2$ strongly depended on types of seed particles. In another example, Cheng et al. (2016) suggested that reactions of $SO_2$ with $NO_2$ in aerosol water could be the source of missing sulfate given the neutralized feature of haze aerosols (with a pH value of about 6 estimated with the ISORROPIA-II thermodynamic equilibrium model). Guo et al. (2017b) argued that the ISORROPIA-II calculation in Cheng et al. (2016) overestimated aerosol water pH, and that transition-metal-catalyzed $O_2$

instead of $NO_2$ was the key oxidant of $SO_2$.

Therefore, solution to the missing sulfate problem (discrepancy between observations and model results) in northern China winter haze remains challenging and controversial. The term sulfate in the atmospheric chemistry literature commonly refers



to inorganic sulfate species. In addition to inorganic sulfate, organosulfur compounds (OS) have also been demonstrated to be present in atmospheric aerosols, including organosulfates ($ROSO_3^-$), sulfones ($RSO_2R'$), and sulfonates ($RSO_3^-$) such as methanesulfonate ($CH_3SO_3^-$, the deprotonated anion of methanesulfonic acid, MSA) and hydroxyalkylsulfonates ($RCH(OH)SO_3^-$) (Eatough and Hansen, 1984; Dixon and Aasen, 1999; Surratt et al., 2008; Tolocka and Turpin, 2012;

Sorooshian et al., 2015). OS may have been misidentified as inorganic sulfate in previous ambient measurements, thus leading to a positive observational bias (as described later). The formation of OS is typically not included in air quality model simulations, and thus can partly explain the missing sulfate problem if concentrations of these species are appreciable. However, OS concentrations in northern China winter haze aerosols have rarely been reported, and their formation mechanisms are also unknown.

In this study, we interpreted measurement data collected by a high-resolution time-of-flight aerosol mass spectrometer (HR-AMS) and a single particle aerosol mass spectrometer (SPAMS) in Beijing winter. We demonstrated the possible presence of OS in haze aerosols and discussed the potential of different OS species. Hydroxymethanesulfonate (HMS, $CH_2(OH)SO_3^-$) was found likely to be the primary OS component. We found that heterogeneous production rate of HMS through reaction of

formaldehyde (HCHO) and $SO_2$ was fast enough to account for the identified OS by HR-AMS. Furthermore, we hypothesized that HMS might lead to additional sulfate production through a mechanism involving aqueous hydroxyl radicals (OH). Finally, we discussed the implications of this heterogeneous HMS chemical mechanism for the missing sulfate problem and also future research needs.

## 2 Methods

### 2.1 Field measurements

Aerosol and gaseous pollutants were measured in urban Beijing during winter 2014 (mid-November to mid-December) (Fig. S1). Chemical composition of non-refractory $PM_1$ was measured by a HR-AMS (Aerodyne Research, Inc., USA). Individual particles (0.2–2 μm) were detected by a SPAMS (Hexin Analytical Instrument Co., China). The Gas and Aerosol Collector Ion Chromatography (GAC-IC) system determined the concentrations of semi-volatile gases ($HNO_3$, $NH_3$, and HCl), and

commercial analyzers were used to measure black carbon (AE33, Magee Scientific, USA), gaseous HCHO (AL4021, Aero-Laser GmbH, Germany), and $SO_2$ and $O_3$ (Thermo Fisher Scientific Inc., USA). Meteorological data were also recorded. Details are provided below and in Table S1.

### 2.1.1 HR-AMS measurements

Particles were vaporized by impaction on a heated surface (600 °C) and the resulting vapors were ionized by an electron impact

ionization source (70 eV) (DeCarlo et al., 2006; Sun et al., 2016). The positive fragment ions generated were detected then using time-of-flight mass spectrometry. The HR-AMS switched every five minutes between the mass sensitive V-mode (mass



resolution of ~2000) and the high resolution W-mode (mass resolution of ~4500). A default collection efficiency of 0.5 was assumed (Middlebrook et al., 2012), because the mass fraction of ammonium nitrate was below 40%, the aerosol particles were moderately acidic (Song et al., 2018), and a silica gel dryer was used to reduce RH in the sampling line. The ionization efficiency calibrations were performed using pure ammonium nitrate particles following Jayne et al. (2000), and the default relative ionization efficiencies (RIE), except for ammonium that was calibrated by ammonium nitrate, were applied to all the chemical species for mass quantifications. It is noted that the calibrated RIE values of sulfate later in 2017 and 2018 ranged from 1.22 to 1.39 (unpublished data), implying a possible overestimation of sulfate mass concentrations by 2%–14%. Mass concentrations of chemical components (ammonium, sulfate, nitrate, chloride, and organics) and inorganic and organic sulfur-containing fragment ions ($H_ySO_x^+$ and $C_xH_yO_zS^+$) were quantified with the standard data analysis software packages (Sueper and collaborators, 2018). These sulfur-containing ions were well separated from adjacent peaks in the observed mass spectra and thus quantified with high confidence (examples in Fig. S2). The uncertainty quantification of chemical species followed Bahreini et al. (2009) (details in Text S1).

### 2.1.2 SPAMS measurements

The SPAMS (Li et al., 2011) was based on the same principle as the ATOFMS (aerosol time-of-flight mass spectrometer) designed by Prather et al. (1994). Ambient aerosol particles were introduced and focused into a narrow beam using an aerodynamic lens. Particles passed through two continuous 532 nm Nd:YAG lasers with velocities determined on the basis of observed travelling times. The individual particles were ionized using a 266 nm Nd:YAG laser, and both positive and negative ions were generated and detected by bipolar time-of-flight mass spectrometry. The Computational Continuation Core software framework based on MATLAB (The MathWorks, Inc., USA) was used to analyze these ions. The negative ion peak at *m/z* −111 has been assigned to HMS ($CH_2(OH)SO_3^-$) with no significant interference from other species in previous laboratory and field studies (Lee et al., 2003; Whiteaker and Prather, 2003; Dall'Osto et al., 2009; Zhang et al., 2012). HMS-containing particles were identified by the presence of a peak at *m/z* −111 with the absolute and relative peak areas greater than 50 and 0.5%, respectively. The negative ion peaks at *m/z* −155, −187, −199, and −215 were also analyzed in order to detect individual organosulfate species (Hatch et al., 2011a).

### 2.1.3 HCHO measurements

The AL4021 analyzer is based on the Hantzsch reaction (HCHO reacts with acetylacetone and ammonia in aqueous solution to form α-α'-dimethyl-β-β'-diacetyl-pyridine which is excited at 400 nm and fluoresces at 510 nm). The Hantzsch reagents were prepared every three days and were kept cool in a refrigerator. A Teflon filter was installed at the sampling inlet to remove particles from the air. This analyzer was calibrated with 1 μM HCHO standard solution every two to three days during the field measurements. The measurement uncertainty is ~20% and the detection limit is ~0.15 ppb (Hak et al., 2005).





### 2.1.4 GAC-IC measurements

Three semi-volatile gases ($NH_3$, $HNO_3$, and $HCl$) were measured with a time resolution of 30 min. The instrument was modified based on the Steam Jet Aerosol Collector (Khlystov et al., 1995) in order to better apply to the heavily polluted conditions in China (Dong et al., 2012). Gases were absorbed in a wet annular denuder and quantified by ion chromatography analyzers. Intercomparison experiments with filter sampling and other online methods showed that the relative uncertainties of GAC-IC were within ± 20% for major species (Song et al., 2018).

### 2.2 Approaches to estimating OS and methanesulfonate with HR-AMS

OS are primarily fragmented into separate organic ions ($C_xH_yO_z^+$) and inorganic sulfur-containing ions ($H_ySO_x^+$) with minimal organic sulfur-containing ions ($C_xH_yO_zS^+$), due to low thermal stability of OS and thermal vaporization and electron ionization of HR-AMS (Farmer et al., 2010; Huang et al., 2015; Hu et al., 2017b). The sulfate concentrations obtained by the standard HR-AMS data analysis are contributed not only by inorganic sulfate but also by OS. $(NH_4)_2SO_4$ is considered the dominant inorganic sulfate species in northern China winter haze aerosols due to the relatively high $NH_3$ levels and the resulting moderate particle acidity (Song et al., 2018). The presence of OS and its contribution to the HR-AMS sulfate (considered as the total sulfate) may be detectable using ratios between different $H_ySO_x^+$, due to different fragmentation patterns of $(NH_4)_2SO_4$ and OS (Hu et al., 2017b). For example, the major $H_ySO_x^+$ from $(NH_4)_2SO_4$ include $S^+$, $SO^+$, $SO_2^+$, $SO_3^+$, $HSO_3^+$, and $H_2SO_4^+$, while some of these ($SO_3^+$, $HSO_3^+$, and $H_2SO_4^+$) are not generated by HMS (Ge et al., 2012; Gilardoni et al., 2016). We observed in Beijing winter that the six ratios of $SO^+/H_ySO_x^+$ and $SO_2^+/H_ySO_x^+$ ($H_ySO_x^+$ refers to $SO_3^+$, $HSO_3^+$, and $H_2SO_4^+$) were highly correlated with each other ($r > 0.9$, $P < 0.001$), and that all of these ratios significantly increased with RH and $PM_1$ concentrations (Fig. 2b and Fig. S3). From the clean and dry conditions (average $PM_1 = 10$ µg m$^{-3}$ and RH = 20%) to the haze (polluted and humid) conditions (average $PM_1 = 160$ µg m$^{-3}$ and RH = 70%), these ion ratios increased by approximately 25–41%. During clean and dry periods, the observed ratios agreed well with values from pure $(NH_4)_2SO_4$ calibrations (Ge et al., 2012; Gilardoni et al., 2016; Hu et al., 2017b), supporting the dominance of $(NH_4)_2SO_4$ over other sulfur-containing compounds. The variations of $SO^+/H_ySO_x^+$ and $SO_2^+/H_ySO_x^+$ were unlikely due to changes in the acidity of haze aerosols (pH of about 4 to about 5 in both clean and haze conditions) (Song et al., 2018) and a clear relationship was not found between these fragment ratios and the fraction of ammonium nitrate in haze aerosols (Chen et al., 2018, Submitted).

The enhancements of $SO^+/H_ySO_x^+$ and $SO_2^+/H_ySO_x^+$ under haze conditions may suggest the presence of additional sulfur compounds, which should either not generate $SO_3^+$, $HSO_3^+$, and $H_2SO_4^+$ ions or have higher ratios of $SO^+/H_ySO_x^+$ and $SO_2^+/H_ySO_x^+$ relative to $(NH_4)_2SO_4$. Since several types of OS (e.g., organosulfates and HMS) satisfy this requirement, the observed HR-AMS mass spectra of inorganic sulfur-containing ions cannot be used to quantify the speciation of OS, but may allow an estimate of sulfate-equivalent OS concentration ($C_{OS}$, µg m$^{-3}$)




$$C_{OS} = M_{SO_4^{2-}} \cdot \left[ \left( SO_{obs}^+ - \overline{R_{cd,SO^+/H_ySO_x^+} \cdot H_ySO_{x,obs}^+} \right)/M_{SO^+} + \left( SO_{2,obs}^+ - \overline{R_{cd,SO_2^+/H_ySO_x^+} \cdot H_ySO_{x,obs}^+} \right)/M_{SO_2^+} \right] \quad (1)$$

where $SO_{obs}^+$, $SO_{2,obs}^+$, and $H_ySO_{x,obs}^+$ ($SO_{3,obs}^+$, $HSO_{3,obs}^+$, and $H_2SO_{4,obs}^+$) are concentrations of the observed inorganic sulfur-containing ions, $M_{SO^+}$, $M_{SO_2^+}$, and $M_{SO_4^{2-}}$ are molar masses, and $R_{cd,SO^+/H_ySO_x^+}$ and $R_{cd,SO_2^+/H_ySO_x^+}$ indicate the average ratios of the two corresponding ions observed during the clean and dry periods. $\overline{R_{cd,SO^+/H_ySO_x^+} \cdot H_ySO_{x,obs}^+}$ is considered the concentration

of $SO^+$ attributable to inorganic sulfate, and its difference from $SO_{obs}^+$ represents the contribution of OS. The same applies to $\overline{R_{cd,SO_2^+/H_ySO_x^+} \cdot H_ySO_{x,obs}^+}$. This approach results in a conservative estimate of $C_{OS}$ because (1) it is assumed that OS do not generate $SO_3^+$, $HSO_3^+$, and $H_2SO_4^+$ ions; and (2) several minor ions (e.g., $S^+$ and $^{33}SO_2^+$) generated by OS fragmentation are not taken into account (Hu et al., 2017a). The uncertainty quantification of $C_{OS}$ is provided in Text S1.

The organic sulfur-containing ions $CH_3SO_2^+$ and $CH_2SO_2^+$ were used as the signature fragments of methanesulfonate (Ge et al., 2012). In our field measurements, an excellent linear correlation ($r = 0.98$, $P < 0.001$) was observed between these two ions and the average ratio of $CH_3SO_2^+$ to $CH_2SO_2^+$ ($2.9 \pm 0.1$) was consistent with the value of $2.9 \pm 0.3$ from the HR-AMS mass spectrum of pure MSA particles found in previous studies (Ge et al., 2012; Huang et al., 2015; Huang et al., 2017), confirming the presence of methanesulfonate in haze aerosols (Fig. S4). We estimated the levels of methanesulfonate using the observed

$CH_3SO_2^+$ and its fraction in the total signal intensity of MSA standards identified in Huang et al. (2017).

**2.3 WRF-Chem air quality model simulation**

The Weather Research and Forecasting model coupled with Chemistry (WRF-Chem; version 3.5.1) (Grell et al., 2005) was adopted to simulate concentrations of particulate sulfate and gas-phase HCHO during the 2014 winter (November–December). Two nested domains were configured to cover East Asia, and model results from the inner domain focused on northern China

with a horizontal resolution of about 27 km were used for analysis. The meteorological initial and boundary conditions were obtained from the NCEP FNL (Final) Operational Global Analysis (NCEP, 2000), and temperature, moisture, and wind fields were nudged to constrain the accuracy of the simulated meteorology. The chemical initial and boundary conditions were provided from MOZART-4 global model simulations of trace gases and aerosols (Emmons et al., 2010). The Lin microphysics scheme (Lin et al., 1983), Rapid Radiative Transfer Model (Mlawer et al., 1997), Goddard shortwave radiation scheme (Kim

and Wang, 2011), Noah Land Surface Model (Chen and Dudhia, 2001), and YSU planetary boundary layer scheme (Hong et al., 2006) were used for the calculations of cloud microphysics, longwave radiation, shortwave radiation, land surface, and boundary layer processes, respectively. The Carbon Bond Mechanism version Z (CBM-Z) (Zaveri and Peters, 1999) gas-phase chemical mechanism coupled with the thermodynamic module MOSAIC (Model for Simulating Aerosol Interactions and Chemistry) (Zaveri et al., 2008) was applied to simulate gas-phase reactions and aerosol processes.




Anthropogenic emissions were obtained from the MIX inventory (Li et al., 2017b) for five sectors, namely power generation, industry, residential, transportation, and agriculture, including the emissions of $SO_2$, $NO_x$, CO, non-methane volatile organic compounds (NMVOCs), $NH_3$, and primary inorganic and organic particulate matters. In addition, biogenic emissions were estimated using the Model of Emissions of Gases and Aerosols from Nature (MEGAN) (Guenther et al., 2006), and open biomass burning emissions were taken from the Global Fire Emissions Database version 4 (GFEDv4) (Randerson et al., 2017). In the WRF-Chem model, sulfate was formed by oxidation of $SO_2$ both in the gas phase (initiated by OH) and in cloud/fog water (by dissolved $H_2O_2$, $O_3$, and transition-metal-catalyzed $O_2$) (Pandis and Seinfeld, 1989)

$$SO_2 + OH + O_2 \rightarrow SO_3 + HO_2 \tag{2}$$

$$HSO_3^- + H_2O_2 \rightarrow SO_4^{2-} + H^+ + H_2O \tag{3}$$

$$SO_{2(aq)} + O_3 \rightarrow SO_4^{2-} + O_2 \tag{4}$$

$$SO_{2(aq)} + \frac{1}{2}O_2 \xrightarrow{Mn^{2+},Fe^{3+}} SO_4^{2-} \tag{5}$$

where $SO_{2(aq)}$ represented the sum of $SO_2 \cdot H_2O$, $HSO_3^-$, and $SO_3^{2-}$. We conducted two simulations (as described in Sect. 3.5): (1) a BASE scenario with normal settings; (2) a 5×EMIS scenario in which primary HCHO emissions from the transportation sector was elevated by a factor of 5.

## 2.4 Apparent heterogeneous sulfate production rate

An apparent heterogeneous uptake of $SO_2$ on aerosols has been shown to be able to compensate for the missing sulfate ($\Delta SO_4^{2-}$) in current air quality models during northern China winter haze episodes (Wang et al., 2014b; Zheng et al., 2015a; Li et al., 2018b). The heterogeneous sulfate production rate, $P_\Delta$, was parameterized with an uptake coefficient $\gamma$ (the fraction of gas-aerosol collisions resulting in chemical reaction) (Jacob, 2000)

$$P_\Delta = \left(R_p/D_g + 4/v\gamma\right)^{-1} S_p[SO_{2(g)}]M_{SO_4^{2-}} \tag{6}$$

where $R_p$ is the average aerosol droplet radius taken as 0.15 µm following Cheng et al. (2016), $D_g$ is the gas-phase diffusion coefficient of $SO_2$, $v$ is the average molecular speed of $SO_2$, $S_p$ is the aerosol surface area per unit volume of air, and $[SO_{2(g)}]$ is the gas-phase concentration. $\gamma$ is $2 \times 10^{-5}$ when RH ≤ 50% and increases linearly from $2 \times 10^{-5}$ to $5 \times 10^{-5}$ when 50% < RH ≤ 100% (Zheng et al., 2015a), consistent with other laboratory and model studies (Wang et al., 2016; Li et al., 2017a). The heterogeneous reaction rate required to produce the identified OS ($P_{OS}$) can be expressed as

$$P_{OS} = \left(C_{OS}/\Delta SO_4^{2-}\right) \cdot P_\Delta \tag{7}$$

## 2.5 ISORROPIA-II thermodynamic equilibrium model calculation

We estimated AWC, aerosol water pH, and ionic strength with the ISORROPIA-II inorganic model (Fountoukis and Nenes, 2007) and also considered the contribution of carbonaceous species. Detailed calculations were given in our recent study (Song et al., 2018). Briefly, ISORROPIA-II predicts the phase partitioning of an $NH_4^+$–$K^+$–$Ca^{2+}$–$Na^+$–$Mg^{2+}$–$SO_4^{2-}$–$NO_3^-$–$Cl^-$–$H_2O$



aerosol and semi-volatile gases (HNO$_3$, NH$_3$, and HCl), and can be used in either forward mode (the total (gas + aerosol) concentration of each species is fixed) or reverse mode (the aerosol concentration of each species is fixed). The forward-mode results were adopted in this study, since the reverse-mode calculations of pH are sensitive to aerosol composition measurement errors and should be avoided. The model inputs were taken from the HR-AMS (bulk PM$_1$ composition) and GAC-IC (semi-

volatile gas) measurements except for crustal species that were estimated based on their levels in PM$_{2.5}$ and typical size distributions (Song et al., 2018). The inorganic aerosol phase state was assumed to be metastable—meaning that the aqueous solution does not crystallize but remains supersaturated when the RH is below the deliquescence RH, although aerosols may reside in a stable state—meaning that the aqueous solution crystallizes once saturation is exceeded (Rood et al., 1989). The assumed phase state led to a small difference in pH (<0.01 unit on average) and an average ~20% difference in AWC (Song et

al., 2018), and thus did not affect the major conclusions of this study. The AWC modeled by ISORROPIA-II has been shown to be in good agreement with those based on ambient measurements in northern China (Wu et al., 2018). The validity of pH calculations was supported by the reasonable agreement between the predicted and observed gas-particle partitioning of semi-volatile species (Song et al., 2018).

The aerosol water associated with carbonaceous species was estimated with the hygroscopicity parameter $\kappa$ (Cheng et al., 2016). $\kappa$ values of 0.06 and 0.04 were used for organics and black carbon, respectively (Song et al., 2018). Contribution of carbonaceous species to AWC was found small (~10%), leading to a minor effect on pH (~0.05 unit). The ionic strength of aerosol water was estimated using the predicted aerosol composition and AWC (Herrmann, 2003). The uncertainty in pH was estimated with a Monte Carlo approach accounting for measurement errors in the model inputs including gas and aerosol

species and meteorological parameters. The 95% confidence interval for calculated pH values was 4.1 to 5.5, very similar to the pH range found in previous northern China winter haze studies (Song et al., 2018).

**2.6 Kinetics and thermodynamics of HMS heterogeneous production**

The chemical mechanism for HMS production can be expressed as (Boyce and Hoffmann, 1984)

$$SO_2 \cdot H_2O \leftrightarrow HSO_3^- + H^+ \tag{8}$$

$$HSO_3^- \leftrightarrow SO_3^{2-} + H^+ \tag{9}$$

$$HCHO + HSO_3^- \overset{k_1}{\leftrightarrow} CH_2(OH)SO_3^- \tag{10}$$

$$HCHO + SO_3^{2-} \overset{k_2}{\leftrightarrow} CH_2(O^-)SO_3^- \tag{11}$$

$$CH_2(OH)SO_3^- \leftrightarrow CH_2(O^-)SO_3^- + H^+ \tag{12}$$

$$CH_2(OH)SO_3H \leftrightarrow CH_2(OH)SO_3^- + H^+ \tag{13}$$

CH$_2$(OH)SO$_3$H (hydroxymethanesulfonic acid, HMSA) dissociates twice to form CH$_2$(OH)SO$_3^-$ and CH$_2$(O$^-$)SO$_3^-$ with the dissociation constants p$K_{a1}$ <0 and p$K_{a2}$ ~12, respectively, and thus, in the pH range of this study, the adduct existed as



$CH_2(OH)SO_3^-$. The acid catalysis of HMS production was significant only at pH < 1 and thus irrelevant in the present context (Olson and Hoffmann, 1989).

Production of HMS in aerosol water was found not limited by the mass transfer processes and hydration of HCHO (Text S2).

The rate for heterogeneous production of HMS, $P_{HMS}$ (in sulfate-equivalent µg m$^{-3}$ h$^{-1}$), was calculated as

$$P_{HMS} = (k_1\alpha_1 + k_2\alpha_2) \cdot [SO_{2(aq)}] \cdot [HCHO_{(aq)}] \cdot AWC \cdot M_{SO_4^{2-}} \qquad (14)$$

where $k_1$ and $k_2$ were, respectively, the forward rate constants for reactions (10) and (11), $[SO_{2(aq)}]$ (defined as the sum of $SO_2 \cdot H_2O$, $HSO_3^-$, and $SO_3^{2-}$) and $[HCHO_{(aq)}]$ were aqueous-phase concentrations estimated with their gas-phase levels and Henry's law constants. $\alpha_1$ and $\alpha_2$ represented the fractions of $HSO_3^-$ and $SO_3^{2-}$ in $SO_{2(aq)}$, respectively, and both were

functions of pH. Production of HMS was reversible and the equilibrium constant $K_{eq}$ could be expressed as

$$K_{eq} = [HMS_{(aq)}]/([HCHO_{(aq)}][SO_{2(aq)}]) \qquad (15)$$

where $[HMS_{(aq)}]$ was the aqueous-phase concentration of HMS. Because the time to reach the equilibrium under winter haze conditions was typically greater than that for haze formation, the precursors and HMS were usually not equilibrated (Text S3). The relevant physical and chemical properties of $SO_2$, HCHO, and HMS are summarized in Tables S2 and S3.

## 3 Results and discussion

### 3.1 Presence of OS in Beijing winter haze aerosols

The standard HR-AMS data analysis usually does not distinguish OS from inorganic sulfate, since OS are fragmented primarily into separate organic ions and inorganic sulfur-containing ions (Hu et al., 2017b). Using the distinct fragmentation patterns of inorganic sulfur-containing ions in the observed HR-AMS mass spectra, we derived a conservative estimate of total OS

concentrations (expressed in sulfate-equivalent µg m$^{-3}$) in PM$_1$ (particles with an aerodynamic diameter below 1 µm), as described in Methods. During several winter haze periods (defined as PM$_1$ > 100 µg m$^{-3}$), OS were estimated to contribute significantly to the total sulfate (TS) identified by the standard HR-AMS data analysis, with an average OS/TS ratio of 17% ± 7% and a maximum ratio of 31% (Fig. 2a–b). Thus, previously reported sulfate concentrations in Beijing winter haze aerosols from HR-AMS measurements may have been biased high due to the presence of OS.

A good positive correlation ($r = 0.82$, $P < 0.001$) was found between OS and the AWC (Fig. 2c) estimated on the basis of the ISORROPIA-II thermodynamic equilibrium model constrained using in situ gas and aerosol compositional and meteorological measurements (see Methods), suggesting that aerosol water serves as a medium enabling production of OS. It has been shown that PM$_1$ are in the liquid phase state during Beijing winter haze periods (Liu et al., 2017b). In contrast, OS were unrelated to

the presence or absence of cloud/fog events (Fig. S5), indicating that the identified OS were less likely formed by cloud/fog



processing (Moch et al., 2017). Although our interpretation of the HR-AMS fragmentation of inorganic sulfur-containing ions cannot directly determine the speciation of OS (organosulfates, sulfones, methanesulfonate, and hydroxyalkylsulfonates), we suggest, through the following analyses, that HMS may be the major OS species in Beijing winter haze aerosols.

### 3.2 Methanesulfonate, sulfones, and organosulfates are likely minor OS species

Contribution of methanesulfonate to OS was only 2%–8% estimated using $CH_3SO_2^+$ as the characteristic fragment in the HR-AMS mass spectra (see Methods and Fig. S4) (Huang et al., 2017). Its estimated concentrations were comparable to those from a previous study (Yuan et al., 2004). The methanesulfonate in Beijing winter is likely formed by oxidation of dimethyl sulfite or dimethyl sulfoxide emitted from waste disposal (Yuan et al., 2004). Aerosol water has been suggested to play important roles in the formation of methanesulfonate and its condensation onto particles (Barnes et al., 2006; Gaston et al., 2010).

Formation of bis-hydroxymethyl sulfone ($C_2H_6SO_4$), the only sulfone that has been identified in ambient aerosols, is inhibited by atmospheric water and is thus unlikely to be important in winter haze (Eatough and Hansen, 1984). Organosulfates, formed through reactions of gaseous organics (e.g., epoxides and aldehydes) and particulate inorganic sulfate, also seem to represent minor contributions to OS based on the following evidence. The most common organosulfates identified previously in ambient

aerosols, such as glycolic acid sulfate ($C_2H_3SO_6^-$), 2-methylglyceric acid sulfate ($C_4H_7SO_7^-$), isoprene epoxydiols sulfate ($C_5H_{11}SO_7^-$), and benzyl (or methyl phenyl) sulfate ($C_7H_7SO_4^-$) (Froyd et al., 2010; Hatch et al., 2011a; Ma et al., 2014), were not detected by SPAMS measurements taken in Beijing winter. Production of organosulfates is enhanced by increased aerosol acidity (Surratt et al., 2010; Hatch et al., 2011b; Riva et al., 2016), whereas the relatively high pH values of about 4 to about 5 for winter haze aerosols in northern China may represent a limiting factor (Song et al., 2018).

### 3.3 HMS is likely the major OS species

Since the only known source of HMS ($CH_2(OH)SO_3^-$) is the nucleophilic addition of $HSO_3^-$ and $SO_3^{2-}$ to HCHO in aqueous solutions (Boyce and Hoffmann, 1984) (Fig. 3a; see Methods), the dominance of HMS in OS can explain the excellent correlation found between OS and AWC (Fig. 2c). The other hydroxyalkylsulfonate species are estimated to be less important than HMS, according to the laboratory experimental data from Hoffmann and colleagues (Olson and Hoffmann, 1989) and the

abundance of different aldehydes in the atmosphere (Text S3).

Mass concentrations of HMS have been observed to be low, on the order of 0.01 μg m$^{-3}$, during several field campaigns in the United States, Germany, and Japan (Dixon and Aasen, 1999; Suzuki et al., 2001; Scheinhardt et al., 2014). However, heterogeneous production of HMS can be fast during winter haze periods in northern China because of moderately acidic pH

of aerosol water, high AWC, high precursor (SO$_2$ and HCHO) concentrations, and low temperature. First, laboratory experiments have indicated that HMS production rates increase rapidly with pH, responding mainly to the dependence of $SO_3^{2-}$



(a more efficient nucleophile than $HSO_3^-$) on pH (Boyce and Hoffmann, 1984; Kok et al., 1986). Values of aerosol water pH of about 4 to 5 obtained during the NCP winter haze events (Song et al., 2018) are 2 to 3 units higher than typically found in North America and Europe (Guo et al., 2017a), implying a factor of $>10^4$ enhancement in HMS production rates. Second, the AWC during winter haze periods (on the order of 100 µg m$^{-3}$) is among the highest in the world (Nguyen et al., 2016),

providing a reactor to facilitate for aqueous reactions. Third, concentrations of both precursor gases are relatively high. Gaseous HCHO has been measured to be about 6 ppb on average in Beijing winter and increases on haze days (Rao et al., 2016). Although emissions of $SO_2$ have rapidly declined, especially since implementation of the National Air Pollution Prevention and Control Action Plan in 2013, its concentrations in the NCP remain much higher than in many other parts of the world (Shao et al., 2018). Last, low temperature in winter increases the solubility of precursor gases in water (Sander, 2015) and thus

enhances HMS production rates. The reaction rate constants for HMS production decrease at low temperature but to a less extent (Boyce and Hoffmann, 1984).

In order to evaluate whether HMS production was fast enough to account for the identified OS, we calculated the rate for HMS production in aerosol water ($P_{HMS}$), and compared this with the apparent heterogeneous reaction rate that would be required

to produce the identified OS ($P_{OS}$). As described in Methods, $P_{HMS}$ calculations involved concentrations of gaseous $SO_2$ and HCHO (Fig. 3c), Henry's law constants, AWC and pH (95% confidence interval: 4.1–5.5), and reaction rate constants from laboratory experiments. HCHO concentrations simulated by the WRF-Chem air quality model (5×EMIS scenario; see Methods and Sect. 3.5) were used because its measurements were available only in December. The modeled HCHO was well correlated with its measured values ($r = 0.8$, $P < 0.001$), but was biased low by ~20% (Fig. 4). $P_{HMS}$ was most sensitive to particle acidity

and increased rapidly with pH, becoming comparable to $P_{OS}$ when using the upper limit of pH (5.5) (Fig. 3d). $P_{HMS}$ calculated here was expected to be conservative. Henry's law constants and kinetic data were obtained in relatively dilute solutions, while aerosol water constitutes a concentrated electrolyte solution. The high ionic strength of aerosol water (~11 M during haze periods) may strongly increase formaldehyde solubility (Toda et al., 2014) and may further enhance kinetic rates for HMS production (Text S4). Gaseous HCHO concentrations used in the calculations were also biased low. It is very likely that the

actual $P_{HMS}$ was comparable to $P_{OS}$ at a pH below 5.5, within the uncertainty range of calculated aerosol water pH. Note that production of HMS was a minor sink of HCHO because the corresponding lifetime of ~4 days was greater than that against photolysis and oxidation by OH (~1 day in Beijing winter).

Our field measurements with SPAMS confirmed the existence of HMS in winter haze aerosols (Fig. 3b). Individual particles

containing HMS were identified by the characteristic mass-to-charge ratio $m/z$ −111 ($CH_2(OH)SO_3^-$) in the SPAMS mass spectra (Whiteaker and Prather, 2003). The observed number concentration of HMS-containing particles ($N_{HMS}$) was closely correlated with AWC ($r = 0.86$, $P < 0.001$), supporting the production of HMS in aerosol droplets. A good relationship was also found between $N_{HMS}$ and the identified OS from HR-AMS (Fig. 3b). Although the SPAMS data showed a significant percentage (~10% during haze periods) of HMS-containing particles in the total particle counts, a quantitative estimate of



HMS mass concentration is not available here because HMS may be fragmented into smaller ions, such as $HSO_3^-$ and $SO_3^-$, depending on the countercations in particles (i.e., matrix effects) (Neubauer et al., 1997; Whiteaker and Prather, 2003). These ions coexist with the HMS peak in the SPAMS mass spectra (Fig. S6).

### 3.4 Implications of heterogeneous HMS chemistry

We have shown above in Beijing winter haze aerosols that a significant fraction of the missing sulfate based on HR-AMS measurements may be attributed to OS, which likely exist primarily as HMS and are misidentified as inorganic sulfate by the standard HR-AMS data analysis. If HMS is assumed to represent the only OS compound, we estimated that it may account for about 1/3 of the missing sulfate in Beijing winter haze aerosols (Fig. S7). Interestingly, HMS would also likely be misidentified as inorganic sulfate ($SO_4^{2-}$) with typical ion chromatography (IC) analysis, another common method used to determine aerosol

chemical compositions. In fact, nearly all of the particulate sulfate measurements in northern China winter haze have been made using these two techniques (He et al., 2014; Cheng et al., 2016; Wang et al., 2016; Li et al., 2017a; He et al., 2018). For anion detection in IC, pH of the carrier fluid (known as eluent, a solution of KOH, NaOH, $NaHCO_3$, etc.) is usually greater than 9 (Wang et al., 2005; Cao et al., 2012; Liu et al., 2016; Han et al., 2017). HMS is unstable under such an alkaline environment and dissociates rapidly into $SO_3^{2-}$ and HCHO (Seinfeld and Pandis, 2016)

$$CH_2(OH)SO_3^- + OH^- \rightarrow HCHO + SO_3^{2-} + H_2O \qquad (16)$$

The characteristic time for HMS dissociation at pH > 9 is less than 1 minute (Seinfeld and Pandis, 2016), much less than the retention time of $SO_4^{2-}$. The product $SO_3^{2-}$ can be rapidly oxidized to $SO_4^{2-}$ by oxidants (e.g., $O_3$ and $H_2O_2$) either generated or dissolved in aqueous extracts.

More intriguingly, heterogeneous HMS chemistry can provide an additional reaction pathway for inorganic sulfate production. Although being resistant to $H_2O_2$ and $O_3$, HMS is oxidized by aqueous OH radicals producing peroxysulfate radicals ($SO_5^{\bullet-}$) (Olson and Fessenden, 1992)

$$CH_2(OH)SO_3^- + OH + O_2 \rightarrow HCHO + SO_5^{\bullet-} + H_2O \qquad (17)$$

$SO_5^{\bullet-}$ is an intermediate in the free-radical chain reactions oxidizing $SO_2$ to $SO_4^{2-}$, and for each attack of OH on HMS, multiple

$SO_4^{2-}$ ions are produced (Fig. S8). Importantly, a HCHO molecule is released by reaction (17), suggesting that this reaction pathway does not result in net consumption of HCHO and that HMS serves as a temporary reservoir of tetravalent sulfur. We speculate from the diurnal patterns of HMS production rates ($P_{HMS}$) and the identified OS concentrations that oxidation of HMS by OH is likely to occur during daytime (Fig. S8). A quantitative rate estimation for this pathway, however, is difficult, because aqueous OH is short-lived and it can be derived as a result of uptake from the gas phase or be generated or scavenged

in the condensed phase (Jacob, 1986; Ervens et al., 2014). It is noted that the gaseous OH levels have been reported to remain relatively high during winter haze periods (Tan et al., 2018).



### 3.5 Future research needs

This study points to a potentially important role of heterogeneous HMS chemistry in explaining the missing sulfate problem during Beijing winter haze episodes. HMS has also been suggested to promote new particle formation by stabilizing sulfuric acid clusters (Li et al., 2018a). Although our field measurements and data interpretation focus on Beijing, this chemical mechanism should be important throughout the NCP because winter haze pollution is regional, as indicated by the distribution of $SO_2$ and HCHO (Fig. S9). Different from many other proposed pathways for sulfate production, HMS chemistry has a characteristic reaction product. More accurate quantification of HMS (e.g., using capillary electrophoresis and ion pairing chromatography (Munger et al., 1986; Scheinhardt et al., 2014)) in future field studies is essential to improve our understanding of this mechanism.

This study reveals the unappreciated role of HCHO in Chinese haze through forming the complex HMS with $SO_2$, and it should be noted that HCHO also serves as a critical source of $HO_x$ (OH + $HO_2$) radicals by photolysis (Rao et al., 2016) and as a carcinogen (Zhu et al., 2017). In spite of its importance, our knowledge of the sources and chemical processes of HCHO during northern China winter haze events remains limited. The standard WRF-Chem model (BASE scenario) underestimated HCHO with a normalized mean bias of −67% (Fig. 4). It is unclear whether primary or secondary sources of HCHO are responsible for its high wintertime levels. Jobson and colleagues have recently suggested that HCHO emissions during motor vehicle cold starts, especially in cold winter, are significantly underestimated in current inventories (Jobson et al., 2017). Accordingly, we conduct a WRF-Chem sensitivity simulation (5×EMIS scenario), which increases primary HCHO emissions from transportation by a factor of 4 and greatly reduces the negative biases in the modeled HCHO (Fig. 4). However, further research should be conducted to elucidate HCHO sources in northern China winter and to design targeted mitigation measures.

### 4 Summary

Combing field measurements and model calculations, we propose a potentially important chemical mechanism, heterogeneous HMS chemistry, for secondary aerosol formation during northern China winter haze episodes. This mechanism involves the production of HMS by HCHO and $SO_2$ in aerosol water, which is favored under northern China winter haze conditions due to several factors including high aerosol water content, moderately acidic pH, high gaseous precursor levels, and low temperature. The produced HMS may be further oxidized to inorganic sulfate through a chain reaction in the presence of aqueous OH in aerosol water. More field, laboratory, and modeling studies are needed in order to elucidate this chemical mechanism and to better understand the emission sources and atmospheric chemical processes impacting HCHO under winter haze conditions.

### Data availability

All data supporting this study are available in this article and its Supplement, or from the corresponding authors upon request.



## Acknowledgments

This work was supported by the Harvard Global Institute and the National Natural Science Foundation of China (91744207, 21607056, 41175114, and 21625701). SPAMS measurements were also funded by the Guangdong Province Public Interest Research and Capacity Building Special Fund (2014B020216005). We thank Jing Cai, Xing Chang, Yunle Chen, Michael

Hoffmann, Lyatt Jaeglé, Lijie Li, Chris P. Nielsen, Viral Shah, Jingyuan Shao, Yu Song, Jay Turner, and Mei Zheng for helpful discussions.

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



**Figures**

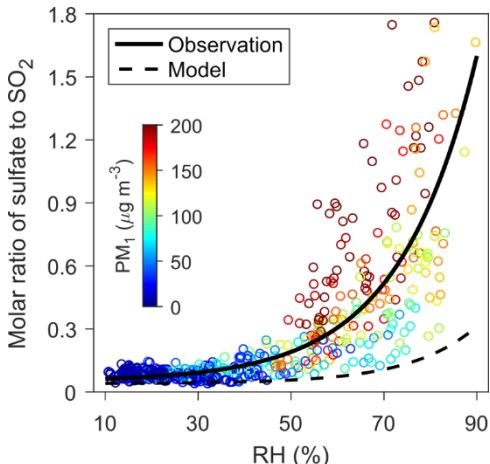

**Figure 1. Relationship between sulfate/SO₂ molar ratio and RH.** The circles indicate hourly observations during 2014 winter in urban Beijing, and are colored according to the observed $PM_1$ (particles with diameter less than 1 µm) concentrations. The solid and dashed curves represent, respectively, the exponential fitting between the observed and modeled sulfate/$SO_2$ ratios and the observed RH, i.e., $y = 0.05 + 7{\times}10^{-3}e^{0.06x}$ ($R^2 = 0.6$) and $y = 0.04 + 5{\times}10^{-4}e^{0.07x}$ ($R^2 = 0.3$). Sulfate concentrations are obtained from HR-AMS $PM_1$ measurements and the corresponding model results are from WRF-Chem simulations.





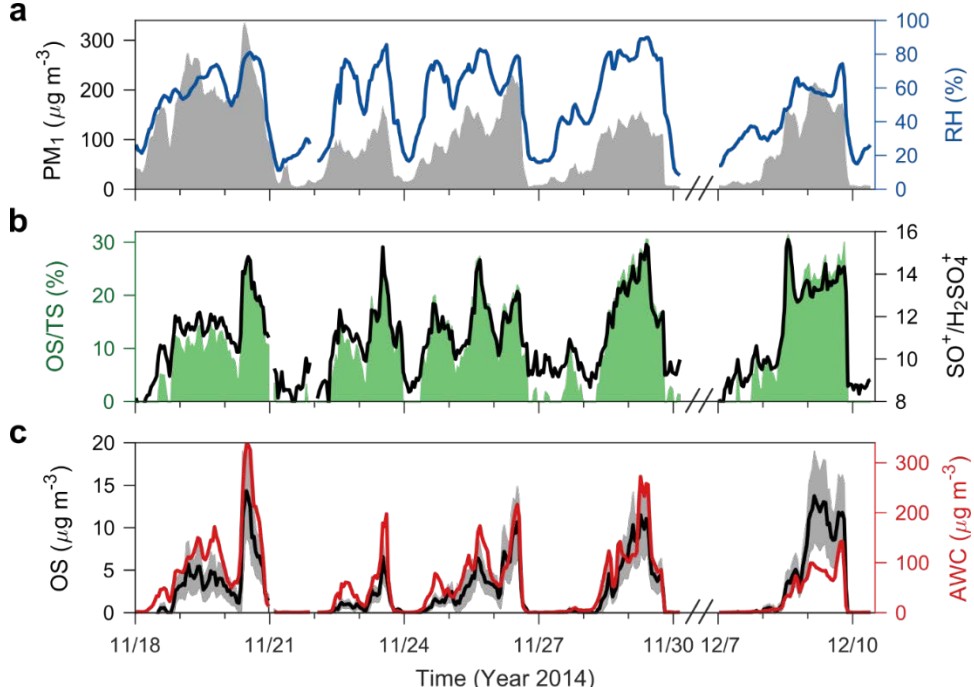

**Figure 2. Possible presence of OS in Beijing winter haze aerosols.** **(a)** $PM_1$ concentrations (left, grey-shaded area) and RH (right, blue line). **(b)** Contributions of OS to TS (left, green-shaded area) and ratios of $SO^+$ to $H_2SO_4^+$ (right, black line). The ratios between the other inorganic sulfur-containing ions are given in Fig. S3. **(c)** Sulfate-equivalent OS concentrations (left, black line) and AWC (right, red line). The grey-shaded area in **(c)** indicates the 1-σ uncertainty range of OS concentrations (on average ~40% during haze periods).



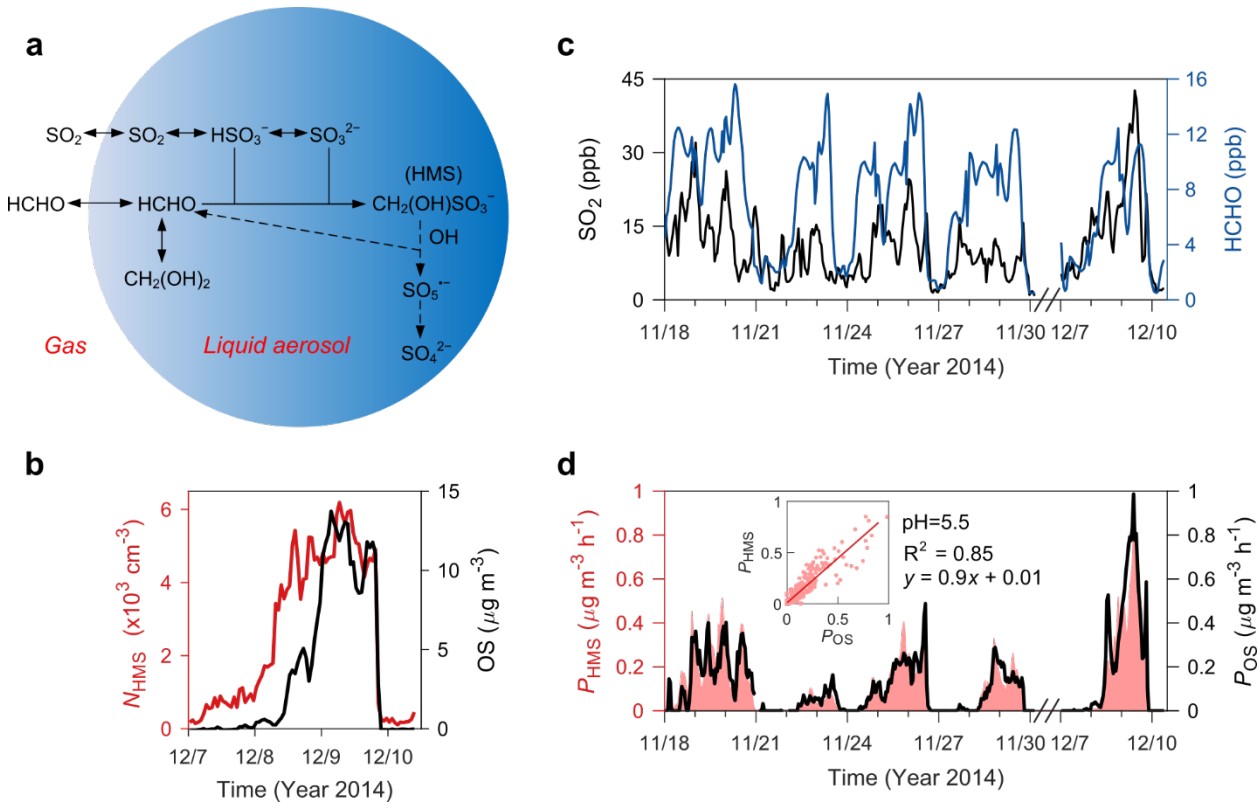

**Figure 3. Production and existence of HMS in Beijing winter haze aerosols. (a)** Schematic of heterogeneous HMS chemistry in northern China winter haze. **(b)** $N_{HMS}$ (left, red line) and OS concentrations (right, black line). $N_{HMS}$ was only available for a haze episode in December 2014 due to instrumentation constraints. **(c)** Gas-phase concentrations of observed $SO_2$ (left, black line) and modeled HCHO (right, blue line). **(d)** $P_{HMS}$ (left, pink-shaded area) and $P_{OS}$ (right, black line). The pink-shaded area shows the uncertainty of $P_{HMS}$ due to the estimated range of pH (4.1–5.5). The inserted figure indicates the linear correlation of $P_{HMS}$ and $P_{OS}$ at a pH of 5.5.





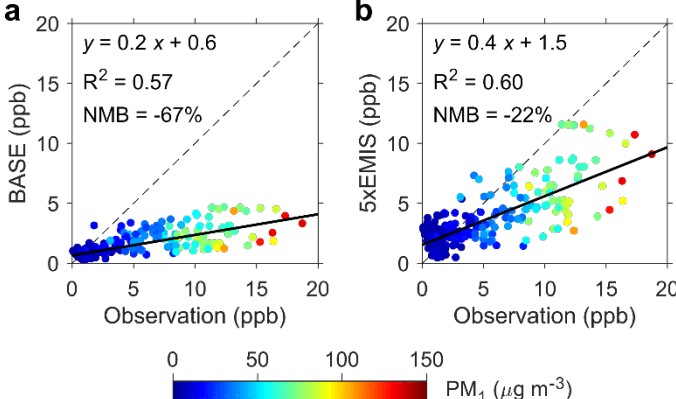

**Figure 4. Comparison of modeled and observed HCHO concentrations.** (a) BASE scenario. (b) 5×EMIS scenario. The dots are colored according to $PM_1$ concentrations. Model results from WRF-Chem are shown.