# Peer review of "Possible heterogeneous chemistry of hydroxymethanesulfonate (HMS) in northern China winter haze"

_Atmospheric Chemistry and Physics, 2018_

## Referee Comment (RC1) · Collett (Referee) · 27 Nov 2018

The authors provide an interesting and comprehensive analysis of the likely contributions of hydroxymethanesulfonate (HMS) to Beijing PM1. This is a nice example of combining fascinating but often forgotten atmospheric chemistry of hydroxyalkylsulfonates from the 1980s and 90s with modern aerosol mass spectrometry techniques. Through the use of HR-AMS, single particle aerosol mass spec, and model simulations, the authors make a compelling case that HMS likely contributes significantly to the sulfur content of Beijing PM1 during humid winter haze conditions.

There are a few points that the authors should consider to improve the manuscript:

[Figure]

1. bottom of p. 9: The authors suggest that a good positive correlation between OS and AWC suggests that aerosol water is key to enabling OS production. One needs to be careful of using correlations to infer causation. My guess is that many Beijing PM1 aerosol species are positively correlated with AWC, both because humid conditions tend to accompany haze events and because AWC depends explicitly on hygroscopic aerosol mass. Is the correlation of OS and AWC higher than for other species (e.g., NH4NO3)?

2. bottom of p. 11: The addition of the SPAMS data, although just for one event when available, greatly adds to the case that HMS is important in this environment. I am puzzled, however, by the observation that the HMS m/z 111 signature was observed in just 10% of particles. If AWC is the main ingredient needed, why isn't HMS contained in most of the particles? I suspect that most Beijing haze particles have substantial AWC in these events. Perhaps differences in pH across particles are important. The authors predict a single pH for PM1 assuming an internal mixture when the aerosol may really be externally mixed. Do the SPAMS data suggest that the HMS occurs in particles of a certain type (e.g., mineral dust particles) that might have a higher pH than other particle types? Are there HR-AMS P-TOF data that can inform us about the HMS size distribution?

3. If there are not aerosol pH-driven differences in HMS production that result in HMS being observed in just a small fraction of particles, perhaps cloud processing is important after all and the HMS was formed in subset of aerosol particles that underwent cloud processing elsewhere in the NCP before being transported to Beijing...?

4. middle of p. 11: It would be helpful if the authors explained and justified their use of a modified HCHO emissions inventory here (or in the Methods section earlier), rather than leaving that explanation to the last page of the manuscript.

5. It would be interesting to look at the competition of aqueous sulfate and HMS formation in Beijing AWC. Both formaldehyde and various oxidants are competing for

dissolved S(IV) in the wet aerosol particles. How do the relative rates of HMS and sulfate production change with aerosol pH and plausible concentrations of reactants and catalysts? The aerosol droplets are small enough and the SO2 likely abundant enough, that the S(IV) oxidation and HMS formation pathways can proceed in parallel, but their relative rates for typical Beijing winter haze conditions would be interesting to outline.

6. The authors do not mention the recent Moch et al. GRL publication, involving several of the same authors that contribute to this paper, that takes a different look at HMS contributions to Beijing PM1. This is likely just a timing issue with submission of the two manuscripts, but should be corrected in production of a revised manuscript.

7. It would be useful for the authors to comment somewhere in the manuscript on the fate of HMS formed in wet haze particles if the RH drops enough for the particles to dry out. Will the HMS still be retained in the dry aerosol?

---

## Short Comment (SC1) · 29 Nov 2018

There is additional indirect evidence of HMS production in China, from retrieved aerosol size distributions inferred from measurements made by sun-sky radiometers in the AERONET and associated networks. In these cases the HMS sized particles were observed only when fog or low altitude layer cloud events were associated with aerosol pollution.

Specifically, two papers have been published regarding this topic, Eck et al. 2012 JGR (see especially Figs 4 and 14) and Li et al. , 2014 in Atmospheric Environment.

Eck, T. F., et al. (2012), Fog- and cloud-induced aerosol modification observed by the Aerosol Robotic Network (AERONET), J. Geophys. Res., 117, D07206, doi: 10.1029/2011JD016839.

Li, Z., Eck, T., Zhang, Y., Zhang, Y., Li, D., Li, L., et al. (2014). Observations of residual submicron fine aerosol particles related to cloud and fog processing during a major pollution event in Beijing. Atmospheric Environment, 86, 187–192. https://doi.org/10.1016/j. atmosenv.2013.12.044

---

## Referee Comment (RC2) · Anonymous Referee #2 · 4 Jan 2019

This manuscript presents very interesting results for the organic sulfate production during haze periods in northern China. Results from this manuscript clearly showed that nearly all the sulfate measured can be attributed to inorganic sulfate during dry and clean periods while up to one third of total sulfate is attributed to organic sulfate (OS). Among them, hydroxymethanesulfonate (HMS) is likely the major OS species. The results are very intriguing and worthy of being further explored. However, several major issues need to be resolved before the manuscript can be publishable.

1. It seems the title is misleading. The major idea of this paper is to conclude that HMS is likely the major OS species as the results and discussion section clearly followed this

logic. In addition, no clear conclusion can be made for rapid sulfate production or even oxidation of HMS which leads to the sulfate formation is still speculative. The chemistry itself is not new and all the reactions in the text were cited from literature. Based on this reason, I would suggest the authors to change the title of this paper to something like "Major contribution of Hydroxymethanesulfonate (HMS) to organic sulfate in northern China winter haze".

2. The authors mentioned that HMS may serve as a reservoir for sulfate, if oxidation of HMS is rapid in the presence of aqueous OH radical, HMS will be quickly converted to sulfate which means that the formation of this intermediate is not important in term of the interpretation of sulfate from the AMS measurements, that is, sulfate from either this pathway or SO2 oxidation is measured as inorganic sulfate. Only when HMS is present in a significant concentration, it becomes important as a major contributor to OS. The authors need to clarify this point.

3. In addition to HMS, the authors list several categories of organic sulfate including methanesulfonate, sulfones, and organosulfates etc. Are there still any other types of OS which might not be considered because of the limitation of the current measurement techniques? In addition, the authors mentioned that several common organosulfates were not detected by the SPAMS; however, that does not mean that they are not significantly present. It is possible that the SPAMS was not capable of detecting them due to its limitation measurement scheme.

4. According to the thermodynamic rules, ammonia will be titrated before it can be taken up by nitrate aerosols. So why it bothers that these fragment ratios should be related to the ammonium nitrate? (line 25 on p5)

Some minor comments

1. Fig. 2b doesn't show any information on the RH. Do you mean Fig. 2a-b?

2. Line 4 on p6 here miss "to be" between "considered" and "the concentration"

[Figure]

3. Line 30 on p9 I don't think you can make cloud related statements since the periods are all covered by cloud

---

## Author Comment (AC1) · 12 Jan 2019

We have addressed peer-referees' comments and put together this authors' response document for *acp-2018-1015*, which includes detailed responses to all the referees and a revised change-tracked manuscript and supplementary material.

**Response to Referee #1**

Comments are in black and responses are in blue.

The authors provide an interesting and comprehensive analysis of the likely contributions of hydroxymethanesulfonate (HMS) to Beijing $PM_1$. This is a nice example of combining fascinating but often forgotten atmospheric chemistry of hydroxyalkylsulfonates from the 1980s and 90s with modern aerosol mass spectrometry techniques. Through the use of HR-AMS, single particle aerosol mass spec, and model simulations, the authors make a compelling case that HMS likely contributes significantly to the sulfur content of Beijing $PM_1$ during humid winter haze conditions. There are a few points that the authors should consider to improve the manuscript:

We thank Dr. Collett for commenting on this manuscript. Our responses to the specific comments and corresponding revisions made in the revised manuscript are provided below:

1. bottom of p. 9: The authors suggest that a good positive correlation between OS and AWC suggests that aerosol water is key to enabling OS production. One needs to be careful of using correlations to infer causation. My guess is that many Beijing $PM_1$ aerosol species are positively correlated with AWC, both because humid conditions tend to accompany haze events and because AWC depends explicitly on hygroscopic aerosol mass. Is the correlation of OS and AWC higher than for other species (e.g., $NH_4NO_3$)?

First, the correlation of OS and AWC ($r = 0.82$) was higher than that of nitrate ($NO_3$) and AWC ($r = 0.69$), as shown in the figure below (*left*: OS vs. AWC, *right*: $NO_3$ vs. AWC). Second, it is also noted from this figure that the intercepts of the linear regressions are different (a small value of 0.19 µg $m^{-3}$ for OS vs. AWC whereas a larger value of 5.8 µg $m^{-3}$ for $NO_3$). This may further imply the causal link between OS and AWC (when there is little AWC, there is little OS). $NO_3$ is known to be formed through different gaseous and heterogeneous pathways (e.g., OH + $NO_2$ and $N_2O_5$ hydrolysis). The correlation between $NO_3$ and AWC could primarily reflect the complex relationships among moisture and physical/chemical evolutions during haze events, as mentioned in the referee's comments.

[Figure]

The small intercept of OS vs. AWC is partly because we used fragmentation patterns of inorganic sulfur ($H_xSO_y^+$) to derive OS levels. The figure below shows the variations of $SO^+$ fragment concentrations and $SO^+/SO_3^+$ fragmentation ratios. We can see that there is a good exponential relationship between $SO^+$ and RH, and that the ratios of $SO^+/SO_3^+$ did not change much under a RH of about 50% but increased quickly above that RH. This critical RH is deliquesced RH value under Beijing winter haze conditions, as predicted in our previous paper using thermodynamic equilibrium analyses (*Song et al. 2018 ACP, Fine-particle pH for Beijing winter haze as inferred from different thermodynamic equilibrium models, doi:10.5194/acp-18-7423-2018*). This may suggest that the existence of OS is associated with wet aerosols.

[Figure]

*Note: the error bars in the right panel show 25% and 75% percentile, and the curve shows the median (50% percentile).*

2. bottom of p. 11: The addition of the SPAMS data, although just for one event when available, greatly adds to the case that HMS is important in this environment. I am puzzled, however, by the observation that the HMS m/z 111

signature was observed in just 10% of particles. If AWC is the main ingredient needed, why isn't HMS contained in most of the particles? I suspect that most Beijing haze particles have substantial AWC in these events. Perhaps differences in pH across particles are important. The authors predict a single pH for PM1 assuming an internal mixture when the aerosol may really be externally mixed. Do the SPAMS data suggest that the HMS occurs in particles of a certain type (e.g., mineral dust particles) that might have a higher pH than other particle types? Are there HR-AMS P-TOF data that can inform us about the HMS size distribution?

The detection of HMS (with the characteristic ion peak at $m/z$ -111) by the SPAMS (using 266 nm Nd:YAG laser for ionization) is subject to matrix effect: HMS ions may be fragmented into smaller ions such as $HSO_3^-$ and $SO_3^-$, depending on the countercations in sampling particles (as described briefly in Sect. 3.4). Several previously published papers (e.g., Neubauer et al., 1997; Whiteaker and Prather, 2003) have discussed this phenomenon. For example, the salt of NaHMS does not produce the $m/z$ -111 peaks at all (fragmented into $SO_2^-$ and $SO_3^-$). The existence of $(NH_4)_2SO_4$ leads to the generation of $m/z$ -111 ion peak, but relative signals ($m/z$ -111:-97 peak area ratio vs. NaHMS: $(NH_4)_2SO_4$) are usually much smaller than unity. For another example, 10% NaHMS in aqueous particles in the mixture of NaHMS and $(NH_4)_2SO_4$ has m/z -111:-97 peak ratios of only 1-3%. The heterogeneity of pH and AWC values among different types of particles is another factor leading to the observed HMS frequency, as the referee has suggested.

We have AMS PToF data in this study as shown below. However, it is very challenging to derive the HMS size distributions from AMS PToF data because fragmentation of HMS and sulfate produces the same $SO^+$ and $SO_2^+$ ions. In the future, it could be possible to estimate the size distributions of HMS by comparing the size differences between $SO^+/SO_2^+$ and $SO_3^+$. As shown below, the average size distribution of sulfate presented a large and broad accumulation mode peaking at ~500 – 600 nm during the severe episode on 9 December (12:00 – 24:00). It is likely that HMS shared the similar size distribution to that of sulfate.

[Figure]

Figure. (a-c) Time series of size distributions of organics, sulfate, and nitrate for the entire study, and (d) average size distributions between 12:00 and 24:00 on 9 December.

3. If there are not aerosol pH-driven differences in HMS production that result in HMS being observed in just a small fraction of particles, perhaps cloud processing is important after all and the HMS was formed in subset of aerosol particles that underwent cloud processing elsewhere in the NCP before being transported to Beijing...?

As described in the earlier comment and response, the fraction of HMS-detected aerosol particles may not necessarily suggest heterogeneity in pH or AWC. But we agree with the referee that HMS may be formed in the subset of aerosol particles that undergo cloud processing before being transported to the observational station. Investigating this hypothesis requires simulations employing a three-dimensional chemical transport model. In the revised manuscript, we have added this hypothesis in the section (Sect. 3.5) discussing future research (see *Page 13 Lines 25-29*):

> "*In addition, three-dimensional chemical transport model studies should be conducted to further explore the HMS formation pathways and the associated uncertainties (e.g., pH values). The modeling simulations may*

*also demonstrate whether significant amount of HMS can be formed in cloud droplets before transported to the ground level".*

We should check whether the model results can reproduce the characteristics observed by mass spectrometry. Given the good relationship found between AWC and OS, we believe aerosol water provides space for the formation of HMS.

4. middle of p. 11: It would be helpful if the authors explained and justified their use of a modified HCHO emissions inventory here (or in the Methods section earlier), rather than leaving that explanation to the last page of the manuscript.

Based on the referee's suggestion, we have moved the explanation for the modified formaldehyde emissions to the Method section. The Results and Discussion section has also been changed accordingly. See revised manuscript *Page 7 Lines 15-23*:

> *"We conducted two model simulations: (1) a BASE scenario with normal settings; (2) a 5×EMIS scenario in which primary HCHO emissions from the transportation sector was elevated by a factor of 5. This is because, as described in Sect. 3 (Results and discussion), the BASE scenario significantly underestimated ambient HCHO concentrations. It remains unclear whether primary or secondary sources of HCHO are responsible for its high wintertime levels. Jobson and colleagues have recently suggested that HCHO emissions during motor vehicle cold starts, especially in cold winter, are significantly underestimated in current inventories (Jobson et al., 2017). Accordingly, a 5×EMIS scenario with increasing primary HCHO emissions from transportation was conducted, which greatly reduces the negative biases in the modeled HCHO and thus was used to calculate the formation rate of HMS."*

5. It would be interesting to look at the competition of aqueous sulfate and HMS formation in Beijing AWC. Both formaldehyde and various oxidants are competing for dissolved S(IV) in the wet aerosol particles. How do the relative rates of HMS and sulfate production change with aerosol pH and plausible concentrations of reactants and catalysts? The aerosol droplets are small enough and the $SO_2$ likely abundant enough, that the S(IV) oxidation and HMS formation pathways can proceed in parallel, but their relative rates for typical Beijing winter haze conditions would be interesting to outline.

We agree with the referee that other chemical pathways consuming $SO_2$ may occur simultaneously in the wet aerosol particles. As described in the Introduction section of our manuscript, potential reactants may include dissolved $NO_2$, transition-metal-catalyzed $O_2$, $H_2O_2$, et al. This manuscript focused on the reaction of HCHO and $SO_2$, and we calculated the time scales of relevant physical and chemical processes. The below figure (Figure S10 in the Supplement) shows that the nucleophilic addition reaction ($\tau_r$) is not limited by mass transfer ($\tau_{dg\&phase}$: gas diffusion and phase equilibrium), by aqueous diffusion ($\tau_{da}$: aqueous diffusion), and by acid hydrolysis ($\tau_{i1}$ and $\tau_{i2}$: two-step dissociation). The fast exchange of $SO_2$ from gas phase to aerosol-water-aqueous phase is primarily a result of small aerosol size, as also suggested by the referee, which also implies that the rates of different heterogeneous pathways are unlikely limited by the supply of $SO_2$.

[Figure]

Quantitative analyses of the relative rates of different $SO_2$ reaction pathways are probably beyond the scope of this particular study. Several previously published papers have addressed several possible pathways and most of them have been referenced in the manuscript (see the Introduction section). A brief summary of these studies was also presented in Introduction:

> "*Their relative importance for sulfate production in winter haze, however, is unknown due to uncertainties in relevant reaction rates and estimates for aerosol water pH values (most reaction pathways are pH-dependent)*".

The findings of this manuscript that HMS contributes up to one thirds of the missing sulfate imply that there are other potential pathways contributing to particulate sulfate. As suggested by the characteristic times in the above figure, mass transfer and acid dissociation of $SO_2$ are unlikely to be the limiting factor under the winter haze conditions.

6. The authors do not mention the recent Moch et al. GRL publication, involving several of the same authors that contribute to this paper, that takes a different look at HMS contributions to Beijing PM1. This is likely just a timing issue with submission of the two manuscripts, but should be corrected in production of a revised manuscript.

In the original manuscript, we cited Moch et al. 2017 poster in AGU 2017 (our paper came out earlier than their GRL paper and ACP journal did not allow cite paper under review). We have changed the citation to Moch et al. 2018 GRL in the revised manuscript. We did not find a relationship between the identified OS and local cloud/fog presence, as described in Sect. 3.1 and Fig. S5.

7. It would be useful for the authors to comment somewhere in the manuscript on the fate of HMS formed in wet haze particles if the RH drops enough for the particles to dry out. Will the HMS still be retained in the dry aerosol?

We have added a brief comment in Sect. 2.6 (*Kinetics and thermodynamics of HMS heterogeneous production*) on the fate of HMS when aerosols dry out. See *Page 9 Lines 10-12*:

> "*It is noted that $CH_2(OH)SO_3^-$ may form salts by the neutralization reaction with ammonium or other cations (e.g., $Na^+$, $K^+$, and $Ca^{2+}$) and undergo precipitation when the ambient RH becomes lower than the efflorescence RH value.*"

**Response to Eck**

Comments are in black and responses are in blue.

There is additional indirect evidence of HMS production in China, from retrieved aerosol size distributions inferred from measurements made by sun-sky radiometers in the AERONET and associated networks. In these cases the HMS sized particles were observed only when fog or low altitude layer cloud events were associated with aerosol pollution. Specifically, two papers have been published regarding this topic, Eck et al. 2012 JGR (see especially Figs 4 and 14) and Li et al. 2014 in Atmospheric Environment.

We thank Mr. Tom Eck very much for commenting on this manuscript. We have added these two references into the revised manuscript:

Eck, T. F., et al. (2012), Fog- and cloud-induced aerosol modification observed by the Aerosol Robotic Network (AERONET), J. Geophys. Res., 117, D07206, doi: 10.1029/2011JD016839.

Li, Z., Eck, T., Zhang, Y., Zhang, Y., Li, D., Li, L., et al. (2014). Observations of residual submicron fine aerosol particles related to cloud and fog processing during a major pollution event in Beijing. Atmospheric Environment, 86, 187–192. https://doi.org/10.1016/j. atmosenv.2013.12.044

Eck et al. (2012) and Li et al. (2014) showed indirect evidence for the existence of HMS from the ground-based remote sensing measurements from the AERONET network. These measurements found in several cases that a fine mode with radius of 0.5 μm might be contributed by HMS, because the associated change was consistent with the observed features In the London fog event (Dall'Osto et al. 2009; cited in our manuscript). Dall'Osto et al. (2009) used the same measurement technique with our study, single particle mass spectrometry, to detect the contribution of HMS to *in situ* aerosol particles. In our cases in Beijing, results of the SPAMS measurements were qualitatively consistent with those from HR-AMS measurements (Fig. 3b). A relationship between HMS signals and local fog events were not observed (Fig. S6 in the revised supplementary).

**Response to Referee #2**

Comments are in black and responses are in blue.

This manuscript presents very interesting results for the organic sulfate production during haze periods in northern China. Results from this manuscript clearly showed that nearly all the sulfate measured can be attributed to inorganic sulfate during dry and clean periods while up to one third of total sulfate is attributed to organic sulfate (OS). Among them, hydroxymethanesulfonate (HMS) is likely the major OS species. The results are very intriguing and worthy of being further explored. However, several major issues need to be resolved before the manuscript can be publishable.

We thank the referee for commenting on this manuscript. Our responses to the specific comments and corresponding revisions made in the revised manuscript are provided below.

1. It seems the title is misleading. The major idea of this paper is to conclude that HMS is likely the major OS species as the results and discussion section clearly followed this logic. In addition, no clear conclusion can be made for rapid sulfate production or even oxidation of HMS which leads to the sulfate formation is still speculative. The chemistry itself is not new and all the reactions in the text were cited from literature. Based on this reason, I would suggest the authors to change the title of this paper to something like "Major contribution of Hydroxymethanesulfonate (HMS) to organic sulfate in northern China winter haze".

We agree with the referee that no clear conclusion is made for the oxidation of HMS to inorganic sulfate. The paper conveys two ideas: one is that HMS is likely a major and significant OS species (from a measurement perspective) and the other is that heterogeneous production of HMS by $SO_2$ and formaldehyde is favored under Beijing winter haze conditions (high aerosol water content, moderately acidic pH, high gaseous precursor levels, and low temperature) (from a modeling perspective). Combining the current knowledge on these two perspectives, we propose the potential importance of heterogeneous hydroxymethanesulfonate (HMS) chemistry in northern China winter haze. Accordingly, the title of the paper has been changed in the revised manuscript to:

"*Possible heterogeneous chemistry of hydroxymethanesulfonate (HMS) in northern China winter haze*"

2. The authors mentioned that HMS may serve as a reservoir for sulfate, if oxidation of HMS is rapid in the presence of aqueous OH radical, HMS will be quickly converted to sulfate which means that the formation of this intermediate is not important in term of the interpretation of sulfate from the AMS measurements, that is, sulfate from either this pathway or $SO_2$ oxidation is measured as inorganic sulfate. Only when HMS is present in a significant concentration, it becomes important as a major contributor to OS. The authors need to clarify this point.

We agree with the referee and have modified relevant sentences in Section 3.4 in order to clarify this point, see *Page 13 Lines 1-17*:

"*HMS should exist as the $CH_2(OH)SO_3^-$ anion in wet aerosols and may form salts with ammonium or other cations when aerosol particles dry out. A possible fate of HMS is to be oxidized by aqueous OH radicals producing peroxysulfate radicals ($SO_5^{\bullet-}$) ...*

*$SO_5^{\bullet-}$ is an intermediate in the free-radical chain reactions oxidizing $SO_2$ to $SO_4^{2-}$, and for each attack of OH on HMS, multiple $SO_4^{2-}$ ions are produced (Fig. S8). Importantly, a HCHO molecule is released by reaction (17), suggesting that this reaction pathway does not result in net consumption of HCHO and that HMS serves as a temporary reservoir of tetravalent sulfur. We speculate from the diurnal patterns of HMS production rates ($P_{HMS}$) and the identified OS concentrations that oxidation of HMS by OH is likely to occur during daytime (Tan et al., 2018) (Fig. S8). But the rate of reaction (17) should be relatively slow when compared with that of reactions (10–11) because a significant level of HMS is expected to reside in the haze aerosol particles. A quantitative rate estimation for this pathway, however, is difficult, because aqueous OH is short-lived and it can be derived as a result of uptake from the gas phase or be generated or scavenged in the condensed phase (Jacob, 1986; Ervens et al., 2014).*"

We note in the above paragraph that the oxidation of HMS by aqueous OH is currently merely a hypothesis/speculation given the previously reported mechanism and the relatively high level of gaseous OH during Beijing winter haze events. Only when HMS is present in a significant concentration, it becomes important as a major contributor to OS. The reaction of HMS and OH cannot be very fast to consume most formed HMS. It may also be important that each attach of HMS by OH has the potential to produce multiple inorganic sulfate.

3. In addition to HMS, the authors list several categories of organic sulfate including methanesulfonate, sulfones, and organosulfates etc. Are there still any other types of OS which might not be considered because of the limitation of the current measurement techniques? In addition, the authors mentioned that several common organosulfates were not detected by the SPAMS; however, that does not mean that they are not significantly present. It is possible that the SPAMS was not capable of detecting them due to its limitation measurement scheme.

This manuscript wrote in the Introduction section that: "*organosulfur compounds ... including organosulfates ($ROSO_3^-$), sulfones ($RSO_2R'$), and sulfonates ($RSO_3^-$) such as methanesulfonate ($CH_3SO_3^-$, the deprotonated anion of methanesulfonic acid, MSA) and hydroxyalkylsulfonates ($RCH(OH)SO_3^-$) (Eatough and Hansen, 1984; Dixon and Aasen, 1999; Surratt et al., 2008; Tolocka and Turpin, 2012; Sorooshian et al., 2015)*". To the best of our knowledge, these above chemical forms include all of the organic sulfur species that have been identified in ambient aerosols. The relative importance of different species varies among different seasons and locations. It should be noted that nitrooxy organosulfates were considered to fall in the scope of organosulfates due to the common sulfate functional group (e.g.,

$C_{10}H_{17}NO_7S$ (proposed structure:  ), a type of secondary organic aerosol that may be formed by α/β-pinene; *Nguyen, Q. T., et al.: Understanding the anthropogenic influence on formation of biogenic secondary organic aerosols in Denmark via analysis of organosulfates and related oxidation products, Atmos. Chem. Phys., 14, 8961-8981, doi:10.5194/acp-14-8961-2014, 2014*).

The single particle mass spectrometry is a common measurement technique to detect the existence of organosulfate compounds, though it is not a suitable way to provide quantitative information for their mass concentrations. Field measurements using such technique have been conducted in many places around the world, for example by Hatch et al. 2011b (cited in our manuscript) in United States and by Wang et al. 2017 (added and cited in the revised manuscript) in China. We have clarified this point in the revised manuscript, see *Page 4 Lines 24-25*:

> *"The negative ion peaks at m/z $-155$, $-187$, $-199$, and $-215$ were also analyzed in order to detect individual organosulfate species. This technique has been employed to measure individual organosulfates in different regions around the world (Hatch et al., 2011a; Wang et al., 2017)."*

4. According to the thermodynamic rules, ammonia will be titrated before it can be taken up by nitrate aerosols. So why it bothers that these fragment ratios should be related to the ammonium nitrate? (line 25 on p5)

The potential influence of ammonium nitrate fraction on the fragmentation patterns of inorganic sulfur is mainly due to the mechanism of AMS measurements. In the preparation of the submitted manuscript, we have asked for and received many useful comments from several experts in the AMS measurements. One of them was that variations of inorganic sulfur fragmentation ratios might be related with the fraction of ammonium nitrate in aerosol particles. There exists a paper currently under peer review covering this issue (we cited this paper in our manuscript):

*Chen, Y., Xu, L., Humphry, T., Hettiyadura, A., Ovadnevaite, J., Huang, S., Poulain, L., Campuzano-Jost, P., Schroder, J., Jimenez, J., Herrmann, H., O'Dowd, C., Stone, E., and Ng, N. L.: Response of the Aerodyne Aerosol Mass Spectrometer to inorganic sulfates and organosulfur compounds: applications in field and laboratory measurements, Environ. Sci. Technol., 2018, Submitted.*

It has been known that the fraction of ammonium nitrate may affect the collection efficiency of AMS from laboratory and field experiments. This effect is related to the phase state of ammonium nitrate, which is a metastable liquid in the atmosphere at any sampling line RH. Middlebrook et al. (2012) (cited in our manuscript) provided a good summary of those experiments. The collection efficiency of AMS generally increases with enhanced fraction of ammonium nitrate. But this known effect should not change the fragmentation patterns of inorganic sulfur.

*Middlebrook, A. M., Bahreini, R., Jimenez, J. L., and Canagaratna, M. R.: Evaluation of composition-dependent collection efficiencies for the Aerodyne aerosol mass spectrometer using field data, Aerosol Sci. Technol., 46, 258-271, doi:10.1080/02786826.2011.620041, 2012.*

Because of the potential influence of ammonium nitrate, we choose to examine the observed relationship between the inorganic sulfur fragmentation patterns and ammonium nitrate fraction. Below is a figure showing this relationship. In the revised manuscript we have added it in the supplementary as *Figure S4*. This figure shows that there was not a clear relationship between ammonium nitrate fraction and the fragmentation ratios of inorganic sulfur from our field measurements. In addition, the fraction of ammonium nitrate was usually lower than 0.3.

[Figure]

*Figure. Relationship between $SO_2^+/H_2SO_4^+$ and fraction of ammonium nitrate. The dots are colored according to $PM_1$ mass concentrations.*

Some minor comments

1. Fig. 2b doesn't show any information on the RH. Do you mean Fig. 2a-b?

Yes. Fig.2a-b respectively show the time series of measured RH and inorganic sulfur fragmentation ratio $SO^+/H_2SO_4^+$. We have changed the corresponding text in Sect. 2.2.

2. Line 4 on p6 here miss "to be" between "considered" and "the concentration"

We have added "to be" in the corresponding sentence in Sect. 2.2. This sentence in the revised manuscript is:

"$\overline{R_{cd,SO^+/H_ySO_x^+} \cdot H_ySO_{x,obs}^+}$ is considered to be the concentration of $SO^+$ attributable to inorganic sulfate"

3. Line 30 on p9 I don't think you can make cloud related statements since the periods are all covered by cloud

Haze periods are typically covered with clear sky. The blue color in the original figure of the supplementary represents clear sky. The original color scheme may be misleading, and thus we have modified the scheme in the supplementary figure, and have also added a legend for different colors, as shown below. Corresponding changes have been made in the revised manuscript.

[Figure]

[revised manuscript text omitted]
_{\mathrm{IE_{NO_3}}}}{\mathrm{IE_{NO_3}}}\right)^2 + \left(\frac{\Delta_{\mathrm{RIE}_X}}{\mathrm{RIE}_X}\right)^2 + \left(\frac{\Delta_{\mathrm{CE}}}{\mathrm{CE}}\right)^2 + \left(\frac{\Delta_{\mathrm{Q}}}{\mathrm{Q}}\right)^2 + \left(\frac{\Delta_{\mathrm{TE}}}{\mathrm{TE}}\right)^2} \tag{S1}$$

where $\frac{\Delta_{\mathrm{IE_{NO_3}}}}{\mathrm{IE_{NO_3}}}$, $\frac{\Delta_{\mathrm{CE}}}{\mathrm{CE}}$, $\frac{\Delta_{\mathrm{Q}}}{\mathrm{Q}}$, and $\frac{\Delta_{\mathrm{TE}}}{\mathrm{TE}}$ represent the relative uncertainties in the nitrate ionization efficiency ($\mathrm{IE_{NO_3}}$), collection efficiency (CE), flow rate (Q), and transmission efficiency (TE), and were estimated to be 10%, 30%, <0.5%, and 10%, respectively. $\frac{\Delta_{\mathrm{RIE}_X}}{\mathrm{RIE}_X}$ represents the uncertainty in the ionization efficiency of the species $X$ relative to nitrate ($\mathrm{RIE}_X$), and depends on the species $X$ (10% for ammonium, 15% for sulfate, and 20% for organics)[1]. Using the above equation, we estimated that the overall relative uncertainties of HR-AMS measurements were 33% (nitrate), 35% (ammonium), 36% (sulfate), and 39% (organics). The uncertainties for chloride were assumed to be 40%.

In order to quantify the uncertainty of sulfate-equivalent organosulfur concentration ($C_{\mathrm{OS}}$), we rewrote Eq. (1) in the main text as

$$C_{\mathrm{OS}} = \frac{M_{\mathrm{SO_4^{2-}}}}{M_{\mathrm{SO^+}}}\left(1 + \frac{M_{\mathrm{SO^+}}}{M_{\mathrm{SO_2^+}}}\frac{\mathrm{SO_{2,OS}^+}}{\mathrm{SO_{OS}^+}}\right) \times \mathrm{SO_{OS}^+} \tag{S2}$$

$$\mathrm{SO_{OS}^+} = \left(1 - \left(\frac{\mathrm{R_{cd,SO^+/SO_3^+}}}{\mathrm{R_{obs,SO^+/SO_3^+}}} + \frac{\mathrm{R_{cd,SO^+/HSO_3^+}}}{\mathrm{R_{obs,SO^+/HSO_3^+}}} + \frac{\mathrm{R_{cd,SO^+/H_2SO_4^+}}}{\mathrm{R_{obs,SO^+/H_2SO_4^+}}}\right)/3\right) \times \mathrm{SO_{obs}^+} \tag{S3}$$

$$\mathrm{SO_{2,OS}^+} = \left(1 - \left(\frac{\mathrm{R_{cd,SO_2^+/SO_3^+}}}{\mathrm{R_{obs,SO_2^+/SO_3^+}}} + \frac{\mathrm{R_{cd,SO_2^+/HSO_3^+}}}{\mathrm{R_{obs,SO_2^+/HSO_3^+}}} + \frac{\mathrm{R_{cd,SO_2^+/H_2SO_4^+}}}{\mathrm{R_{obs,SO_2^+/H_2SO_4^+}}}\right)/3\right) \times \mathrm{SO_{2,obs}^+} \tag{S4}$$

where $\mathrm{SO_{OS}^+}$ and $\mathrm{SO_{2,OS}^+}$ are the concentrations of the $\mathrm{SO^+}$ and $\mathrm{SO_2^+}$ ions attributable to OS, respectively. The observed ion ratios are defined as $\mathrm{R_{obs,SO^+/SO_3^+}} = \frac{\mathrm{SO_{obs}^+}}{\mathrm{SO_{3,obs}^+}}$, $\mathrm{R_{obs,SO^+/HSO_3^+}} = \frac{\mathrm{SO_{obs}^+}}{\mathrm{HSO_{3,obs}^+}}$, $\mathrm{R_{obs,SO^+/H_2SO_4^+}} = \frac{\mathrm{SO_{obs}^+}}{\mathrm{H_2SO_{4,obs}^+}}$, $\mathrm{R_{obs,SO_2^+/SO_3^+}} = \frac{\mathrm{SO_{2,obs}^+}}{\mathrm{SO_{3,obs}^+}}$, $\mathrm{R_{obs,SO_2^+/HSO_3^+}} = \frac{\mathrm{SO_{2,obs}^+}}{\mathrm{HSO_{3,obs}^+}}$, and $\mathrm{R_{obs,SO_2^+/H_2SO_4^+}} = \frac{\mathrm{SO_{2,obs}^+}}{\mathrm{H_2SO_{4,obs}^+}}$. $\mathrm{R_{cd,SO^+/SO_3^+}}$, $\mathrm{R_{cd,SO^+/HSO_3^+}}$, $\mathrm{R_{cd,SO^+/H_2SO_4^+}}$, $\mathrm{R_{cd,SO_2^+/SO_3^+}}$, $\mathrm{R_{cd,SO_2^+/HSO_3^+}}$, and $\mathrm{R_{cd,SO_2^+/H_2SO_4^+}}$ represent the average ratios of the two corresponding ions in the clean and dry periods. We further defined A and B and rewrote Eq. (S2)

$$A = 1 - \left(\frac{\mathrm{R_{cd,SO^+/SO_3^+}}}{\mathrm{R_{obs,SO^+/SO_3^+}}} + \frac{\mathrm{R_{cd,SO^+/HSO_3^+}}}{\mathrm{R_{obs,SO^+/HSO_3^+}}} + \frac{\mathrm{R_{cd,SO^+/H_2SO_4^+}}}{\mathrm{R_{obs,SO^+/H_2SO_4^+}}}\right)/3 \tag{S5}$$

$$B = 1 - \left( \frac{R_{cd,SO_2^+/SO_3^+}}{R_{obs,SO_2^+/SO_3^+}} + \frac{R_{cd,SO_2^+/HSO_3^+}}{R_{obs,SO_2^+/HSO_3^+}} + \frac{R_{cd,SO_2^+/H_2SO_4^+}}{R_{obs,SO_2^+/H_2SO_4^+}} \right) / 3 \tag{S6}$$

$$C_{OS} = \frac{M_{SO_4^{2-}}}{M_{SO^+}} \left( 1 + \frac{M_{SO^+}}{M_{SO_2^+}} \frac{B}{A} R_{obs,SO_2^+/SO^+} \right) \times SO_{obs}^+ A \tag{S7}$$

where $R_{obs,SO_2^+/SO^+} = \frac{SO_{2,obs}^+}{SO_{obs}^+}$. The uncertainty of $C_{OS}$ was calculated by propagating the uncertainties of each term in

Eqs. (S5–S7). The uncertainties of $R_{cd,SO^+/SO_3^+}$, $R_{cd,SO^+/HSO_3^+}$, $R_{cd,SO^+/H_2SO_4^+}$, $R_{cd,SO_2^+/SO_3^+}$, $R_{cd,SO_2^+/HSO_3^+}$, and $R_{cd,SO_2^+/H_2SO_4^+}$

5    were estimated by the standard deviations of the corresponding ion ratios observed during the clean and dry periods.

The uncertainties of $R_{obs,SO_2^+/SO^+}$, $R_{obs,SO^+/SO_3^+}$, $R_{obs,SO^+/HSO_3^+}$, $R_{obs,SO^+/H_2SO_4^+}$, $R_{obs,SO_2^+/SO_3^+}$, $R_{obs,SO_2^+/HSO_3^+}$, and

$R_{obs,SO_2^+/H_2SO_4^+}$ were obtained by the standard deviations of the 5-minute samples in each hour. The relative uncertainty

of $SO_{obs}^+$ was estimated by Eq. (S1) assuming the same RIE as sulfate.

10    **Text S2. Mass transfer of heterogeneous HMS production**

In order to examine whether the production of HMS was limited by the kinetics of mass transport, we estimated the

characteristic time scales ($\tau$) for the HMS chemical reaction, the mass transfer steps (including gas-phase diffusion,

interfacial transport, hydrolysis/ionization, and aqueous-phase diffusion), and formaldehyde hydration. The average

values during winter haze episodes ($PM_1 > 100$ µg m$^{-3}$) were used in our calculation: $[SO_{2(g)}]$ = 14 ppb, $[HCHO_{(g)}]$

15    = 10 ppb, $T$ = 278 K, $PM_1$ = 160 µg m$^{-3}$, and AWC = 100 µg m$^{-3}$. The characteristic times for the chemical loss of

$SO_2$ and HCHO, $\tau_{r,SO_2}$ and $\tau_{r,HCHO}$, were estimated by

$$\tau_{r,SO_2} = \{(k_1\alpha_1 + k_2\alpha_2)[HCHO_{(aq)}]\}^{-1} \text{ and } \tau_{r,HCHO} = \{(k_1\alpha_1 + k_2\alpha_2)[SO_{2(aq)}]\}^{-1} \
[revised manuscript text omitted]